

# From lidar scans to roughness maps for wind resource modeling in forested areas

Rogier Floors[1], Peter Enevoldsen[2,3], Neil Davis[1], Johan Arnqvist[4], and Ebba Dellwik[1]

[1]Department of Wind Energy, Technical University of Denmark
[2]Center for Energy Technologies, Aarhus University, Denmark
[3]Envision Energy, Denmark
[4]Department of Earth Sciences, Uppsala University, Sweden

*Correspondence to:* Rogier Floors (rofl@dtu.dk)

**Abstract.** Applying erroneous roughness lengths can have a large impact on the estimated performance of wind turbines, particularly in forested areas. In this study, a new method called the Objective Roughness Approach (ORA), which converts tree height maps created using airborne lidar scans to roughness maps suitable for wind modeling, is evaluated via cross-predictions between different anemometers at a complex forested site with seven tall meteorological masts using the Wind

Atlas Analysis and Application program (WAsP). The cross-predictions were made using ORA maps created at four spatial resolutions and from four freely available roughness maps based on land-use classifications. The validation showed that the use of ORA maps resulted in a closer agreement with observational data for all investigated resolutions compared to the land-use maps. Further, when using the ORA maps, the risk of making large errors ($> 25$ %) in predicted power density was reduced by 40–50 % compared to satellite based products with the same resolution. The results could be further improved for high-

resolution ORA maps by adding the displacement height. The improvement when using the ORA maps came down to two factors, first they had a higher roughness length for forests, which was confirmed to by increasing the forest roughness value of the land-use based maps to the value of the ORA map, and second, due to the higher resolution of the ORA data, since the ORA maps with the highest resolution had the largest reduction in mean absolute errors.

*Copyright statement.* TEXT

# 1   Introduction

In the past decade, there has been a significant increase in installed wind power capacity in forested areas across Northern Europe (Enevoldsen and Valentine, 2016). This has been made possible due to the increased hub heights and improved technologies of modern turbines (Enevoldsen, 2016). However, these forested sites typically represent a challenging environment for wind turbines because of high turbulence and wind shear levels (Arnqvist et al., 2015). The forested landscape is rarely

homogeneous, containing both forest edges and roughness changes that affect shear and turbulence, and can produce sharp gradients in the flow (*e.g* Poëtte et al., 2017). This, in turn, makes the accurate siting of wind turbines in forested areas critical.



For all of these reasons, it is important to parametrize forest effects correctly in wind models aimed at the application of wind turbine siting. The aim of the current study is twofold: (1) to introduce a new and automated way of using highly detailed forest information with the WAsP method (Troen and Petersen, 1989) and (2) to investigate how prediction errors change when using different maps to represent the forest.

Starting with the second aim, a short background for the research in forest wind meteorology is given. In surface layer theory, the effect of a homogeneous forested area on the wind speed $U$ is parametrized through a suitable value for the roughness length $z_0$ as well as a displacement height $d$ in the well-known logarithmic wind profile:

$$U = \frac{u_*}{\kappa} \left[ \ln \frac{z-d}{z_0} - \Psi_m \right], \tag{1}$$

where $u_*$ is the friction velocity, $z$ is the height above ground and $\Psi_m$ represents the stability correction to the profile and
depends on $z/L$, where $L$ is the Monin-Obukhov length (Businger et al., 1971). Based on this equation, experimental studies on the wind profile over forested areas typically present estimates of $z_0$ and $d$ as a pair (*e.g.* Thom, 1971; Hicks et al., 1975; Molder et al., 1999; Dellwik and Jensen, 2005; Arnqvist et al., 2015). A dependence between measurable forest characteristics (height and density) and $z_0$ and $d$ was suggested by Jackson (1981); Raupach (1994). According to Raupach (1994), a dense forest should be represented by a relatively lower $z_0$ and a higher $d$ than a sparser forest with the same tree heights. For sites
without forest density information, Garrat (1992) recommended setting $z_0 = 0.1h$ and $d = 2/3h$, where $h$ is the tree height.

The WAsP method (Troen and Petersen, 1989), which allows for the prediction of a wind climate at a new location using an observed wind climate, includes $z_0$ in two different ways. In the absence of nearby obstacles, the difference between the predicted and observed wind climate is calculated through a *geostrophic roughness* model that uses Eq. 1 in combination with the geostrophic drag law for vertical and horizontal extrapolation. Heterogeneity in roughness and terrain height is modelled
through a *roughness change* model, which consists of an internal boundary layer model that is driven by lines of roughness change lines on a roughness map, and a flow-over-terrain model that simulates the speed-up effects due to changes in terrain height (Troen and De Baas, 1987).

Using data from small beech forests, Dellwik and Jensen (2000) showed that the internal boundary layer growth closely resembled the prediction of the WAsP roughness change model. In a later study, (Dellwik et al., 2006), it was demonstrated
that WAsP could correctly predict the wind climate at the forest sites from observations at a nearby agricultural area, when the $z_0$ value was set to the values found in the surface layer studies mentioned above, which were significantly higher than those that are typically used for forest studies. However, these studies investigated sites with near-surface wind observations in an open landscape with small distinct forests, which differ greatly from the Northern European sites that have recently started to be developed for wind energy. The Northern European sites have a landscape that is dominated by forests with smaller areas
of lower-roughness clearings and lakes, leading to study areas where both the observation and predicted sites are located in a high roughness forested area. This study looks to determine if the roughness map has as much impact at such sites, as was demonstrated for the sites in Dellwik et al. (2006).

There is currently no consensus on how to take $d$ in Eq. (1) into account in WAsP (Enevoldsen, 2017). Many siting engineers use terrain height products from the Space Radar Topography Mission (SRTM) to describe the orography effects in WAsP





and WindPRO, which includes the vegetation height in the terrain height, meaning that for forested areas, the displacement height is already included to some extent (Kellndorfer et al., 2004; Crockford and Hui, 2007). In Dellwik et al. (2006), a terrain height map that did not include the forest height was used, therefore, the displacement height was included by adjusting the height of the wind observations as a post-processing step. However, this post-processing approach does not work close to

forest edges, since they cause the flow to speed up similar to a terrain height escarpment (Dellwik et al., 2014). In forested sites, measurement masts are often placed in clearings near forest edges, suggesting that the displacement height is particularly important. This study includes an investigation into the impact of adding $d$ to the terrain data on the model predictions.

The performance of the flow modeling depends greatly on the availability of accurate maps of terrain height and $z_0$. It is common for siting-engineers to use $z_0$ maps derived from land-use classifications, and many siting tools, such as WindPRO,

which uses WAsP as the flow model, provide these maps for download directly in the software. These maps are quite useful in that they can be obtained anywhere in the world, and do not require additional time to locate and implement in the siting tool. However, because, the land-use categories were not developed for wind energy applications, a specific land use class may not correspond to a unique effect on the wind field. Additionally, neither the forest height nor forest density are typically part of land-use classifications making it difficult to apply the relationship suggested by, *e.g.*, Raupach (1994) to forested areas.

This information may be found either by a site visit, or from the forest owners, but it is difficult to find consistent high-quality information over the large areas required for wind resource assessment. This difficulty was one of the starting points for this study and is connected with aim (1) stated above.

An attractive solution to obtaining high-quality information about the forest characteristics can be found in the raw data from remote-sensing Airborne Laser Scans (ALS) for land surface mapping (Clark et al., 2004). During the last decade, the ALS

technology has seen a rapid expansion and ALS mapping campaigns have been performed for entire countries (*e.g.* Nilsson et al., 2017). Simultaneously, the price of ALS studies has decreased, making mapping campaigns for in-depth studies possible for both research and commercial wind projects (*e.g.* Boudreault et al., 2017). The raw data from the scans is stored in standard format, allowing siting engineers to use the same data source for land surface description for different sites (American Society for Photogrammetry & Remote Sensing, 2010). Boudreault et al. (2015) introduced a method that used the raw data from ALS

campaigns in canopy-resolving flow models. Here, we translate the derived forest height from ALS data to a roughness map for use in a WAsP analysis. The ALS based model results are compared with model results based on the standard land-use based maps. The four land-use based maps tested in this study had different resolutions, and included a varying level of detail. To investigate the importance of resolution, the roughness maps made from the ALS scans were created at resolutions matching the land-use models. In addition to evaluating model performance, the different resolutions allowed for the investigation of

how the WAsP handled the large amount of data, with a particular focus on retaining the speed of the WAsP calculation.

This study performs validation and analysis of the model results for the cross-prediction of 7 masts located in a predominantly forested area in central Sweden. At such high latitudes, icing on cup anemometers and wind vanes is a common problem for wind measurement (Bredesen et al., 2017). There were periods with frozen cup anemometers and wind vanes: the process for removal of these periods is described in (Sec. 2). The processing of the ALS data and its conversion to roughness length and

displacement height is presented in Sec. 3. Sec. 3 also contains descriptions of the land-use classification based datasets. The





WAsP flow model and the cross-prediction method used for the model assessment are then described in Section 4. Section 5 presents the results of the comparison between model performance for the different roughness products. Finally, we discuss uncertainties, opportunities and possible improvements in Section 6.

## 2 Wind measurements

### 2.1 Site and instrumentation

The site used in this study is located on two forested ridges in central Sweden (Fig. 1, left), approximately 140 km from the Baltic sea coast and approximately $3°$ below the polar circle. Within an area of $10 \times 10$ km, the site had two 59 m and five 100 m tall meteorological masts, each of which had cup anemometers and wind vanes at several heights (Table 1). The measurement periods were different from mast to mast, but all masts were operational between February $23^{rd}$ 2009 and February $18^{th}$ 2010. Three different types of cup anemometers were used; NRG #40 anemometers, manufactured by NRG Systems, Inc., were used for the profile measurements on all masts, the two 59 m masts each had two top mounted WindSensor (Risø) P2546A anemometers (Pedersen, 2004), and the top mounted anemometers on the 100 m masts were Thies First Class anemometers.

**Table 1.** Measurement period and instrumentation for each of the seven masts.

| Mast No | Meas. period | Top | Profile | Heights (m) |
|:---:|:---|:---|:---|:---:|
| 1 | Jan. 2008 - Mar. 2010 | Risø P2546A | NRG40 | 59.0, 59.0, 57.0, 44.5, 31.5 |
| 2 | Dec. 2007 - Feb. 2010 | Risø P2546A | NRG40 | 59.0, 59.0, 57.3, 44.0, 32.1 |
| 3 | Nov. 2008 - Feb. 2010 | Thies First Class | NRG40 | 100.7, 100.7, 96.4, 80.7, 57.8 |
| 4 | Feb. 2009 - Feb. 2010 | Thies First Class | NRG40 | 100.8, 100.8,96.4, 80.8, 57.7 |
| 5 | Feb. 2009 - Nov. 2011 | Thies First Class | NRG40 | 100.8, 100.8,96.4, 80.9, 57.8 |
| 6 | Feb. 2009 - Feb. 2010 | Thies First Class | NRG40 | 100.8, 100.8,96.4, 80.7, 57.6 |
| 7 | Feb. 2009 - Jun. 2011 | Thies First Class | NRG40 | 100.8, 100.8,96.4, 80.9, 57.8 |

### 2.2 Treatment of observational wind data

An initial screening of the observed wind data (10 min averages) showed significant inconsistencies in both wind speed and wind direction. The erroneous data were prevalent during winter, and most likely caused by ice growth on the instruments. During most of these periods, the wind speed would have a constant near-zero value, signifying that the cup was completely frozen. However, there were also times where the wind speed of a particular anemometer was much lower than expected given other wind measurements, suggesting that the anemometer was able to turn, but at a lower rate than if ice free. The following data screening steps were applied to clean the data of ice contaminated measurements:



1. Removal of periods when the cup anemometer was clearly malfunctioning; requiring that $0 \leq U \leq 50$ m s$^{-1}$ and $I_u < 1$, where $U$ is the wind speed and $I_u = \sigma_u/U$ is the turbulence intensity, and $\sigma_u$ is the standard deviation of the wind speed.

2. Removal of periods with constant wind speed; requiring $U_i \neq U_{i\pm1}$, where $i$ denotes a 10 minute block average.

3. Removal of ice-affected data by comparing pairs of cup anemometers in each mast; requiring the relative difference to at least three other anemometers on the same mast were within $3\sigma$ of the ice-free mean relative difference, where there relative difference is defined as $RD = (U_i - U_j)/U_i$ is the relative difference, $\sigma$ is the standard deviation of $RD$, and the ice-free period is defined as May to September.

Data passing these criteria were associated with a quality control value $(QC)$ of 1. These steps removed between 8 and 26 %
of the data.

In addition to the ice related issues, it was found that the NRG cup anemometer wind speeds were systematically lower than the top anemometers. We attribute this difference to both mast shadowing and a systematic small instrumental error, possibly related to the temperature response of the instrument. Finally, there was a slight offset between the two top anemometers related to wind direction. To get an accurate wind profile, both the NRG offset error and the top wind speed measurement were
corrected as follows:

1. Determine which top cup anemometer data to use based on wind direction.

2. Calculate the expected top wind speed from the profile cups by linear extrapolation from the two highest profile cups.

3. Define a correction factor as the ratio between the actual top cup measured wind speed and the profile estimated value for each 10-minute period.

4. Apply the correction factor to all wind speeds in the profile.

The final data cleaning step required that all cup anemometers had simultaneous values of $QC = 1$ to ensure that the wind distributions used in the cross-predictions were generated from the same period across all anemometers. After applying the filtering steps, 8764 ten minute mean wind speeds were available at each height and mast, which corresponds to approximately 1.5 months of data. The effect of the data filtering is shown in Sec. 5.
After cleaning the data, an 'observed wind climate' was created for use in WAsP. The 'observed wind climate' is a histogram of wind speeds for different wind direction sectors, i.e. the frequency of the wind for each wind speed and direction bin. For this study, a data discretization of 1 m s$^{-1}$ for wind speed and 30°for wind direction were used.



## 3   Site land cover description

### 3.1   Roughness maps based on land use classes

A common approach of obtaining roughness information for wind flow modeling is through the conversion of land-use classifications, typically derived from satellite data to a roughness length through the use of look-up tables. In this study, four

different sources of land use classifications are investigated (Table 2). These four datasets are provided for download by EMD International A/S through their WindPRO software, which is widely used for wind resource assessments.

**Table 2.** Summary of the different data sources that were used for creating the maps

| Name | Abbrevation | Spatial Resolution (m) | Number of classifications | Satellite Coverage Date | Reference |
|---|---|---|---|---|---|
| Corine land cover | CORINE100 | 100 | 44 | 2006 | EEA (2007) |
| ESA GLOBCOVER | GLOB300 | 300 | 23 | 2009 | Bontemps et al. (2011) |
| Modis Vegetation Continuous Field | MODIS500 | 500 | 7 | 2001 | DiMiceli et al. (2011) |
| Global Land Cover Classification (GLCC) | GLCC1000 | 1000 | 24 | 1992-1993 | Hansen et al. (2000) |

All four of the land-use datasets used in this study are derived from satellite measurements, however, there are some significant differences. The CORINE dataset has the highest spatial resolution at 100m, but only covers Europe. This dataset has frequqently been used in mesoscale meteorological modeling (Pindea et al., 2002), due to the high spatial resolution and large

number of classification classes. Five of the 44 CORINE classes are associated with forests. The GLOBCOVER dataset has been used in the Global Wind Atlas (Mortensen et al., 2017), and while it has a lower resolution than the CORINE data, it is a more recent dataset, provides global coverage, and, important for this study, has 8 forest classes out of the 23 different land cover classifications. The MODIS Vegetation Continuous Field is not a true land-use dataset. Instead it provides information about three components of the ground cover: percentage of tree cover, percentage of non-tree vegetation, and percentage of

bare ground (Carroll et al., 2010). Finally, the Global Land Cover Characterization (GLCC) dataset provides a coarse 1000 m resolution land cover dataset, and is also the oldest of the datasets.

When downloading the data using WindPro, standard roughness conversion tables are used. The values for classes in our model domain are found in table 3. While some of the datasets have more forest classes than others, when translated to roughness there are at most two different values of roughness length for the different forest types. Additionally, the values for

the different roughness classes are relatively similar across the different products, with the exception of CORINE, which has a category of peat bogs with very low roughness that is not in the other datasets. It is also important to note that the roughness lengths in these tables are not related to the local forest height, just the generic class description.



**Table 3.** Table with the name of the land use category and the prescribed value of roughness length, for each land-use dataset.

| Land Use Definition | Equivalent Roughness Length (m) |
|---|---|
| **CORINE** | |
| Coniferous forest | 0.5 |
| Transitional woodland-shrub | 0.4 |
| Mixed forest | 0.5 |
| Agriculture | 0.056 |
| Lake | 0 |
| Peat bogs | 0.0184 |
| Complex cultivation | 0.0560 |
| Pastures | 0.0360 |
| River | 0 |
| Non-irrigated arable land | 0.0560 |
| Burnt areas | 0.2 |
| Broad-leaved forest | 0.5000 |
| **MODIS VCF** | |
| Canopy cover = 51-75 % | 0.4000 |
| Canopy cover = 26-50 % | 0.3031 |
| Canopy cover = 1-25 % | 0.1000 |
| Canopy cover = 76-100 % | 0.5278 |
| Inland water | 0.0005 |
| Canopy cover = 0 % | 0.0382 |
| **GLOBCOVER** | |
| Closed to pen mixed broadleaved and needle-leaved forest | 0.4000 |
| Open needleleaved deciduous or evergreen forest | 0.3000 |
| Sparse vegetation | 0.0500 |
| Mosaic grassland | 0.3000 |
| Closed broadleaved deciduous forest | 0.4000 |
| Closed to open grassland or woody vegetation on regularly flooded | 0.2000 |
| Water bodies | 0 |
| Mosaic forest or shrubland | 0.4000 |
| Closed needleleaved evergreen forest | 0.5000 |
| **Global Land Cover Classification** | |
| Evergreen needleleaf Forest | 0.5000 |
| Water bodies | 0.0002 |
| Cropland | 0.0700 |
| Dryland cropland | 0.1000 |
| Mixed forest | 0.4000 |





## 3.2 Airborne Laser Scans

Although using land-use maps is attractive due to its simplicity, the subjective conversion of land-use classes to roughness length can significantly affect the uncertainty of wind resource estimations, particularly for forested sites (Kelly and Jørgensen, 2017). Therefore, a more objective approach is presented in this section.

### 3.2.1 Estimating elevation and forest height

The area at and around the site was scanned by airborne laser in 2013 from June to October, with a small part of the northern regions being collected in 2012. This timing is fortunate, as the scan period is quite close to the observational period of this study. The scans were made as part of a national survey (Lantmäteriet, 2016). Around 50 GB of data in the .las format, covering a 40x40 km area, were downloaded from the Swedish national database for this study. The reflection points in the point cloud were classified into ground, water, and vegetation points, by the data provider. The data were first processed from the ALS point cloud to a terrain height map, a forest height map, and a map of all the water areas (lakes and rivers) at a $20 \times 20$ m$^2$ grid resolution, using the approach of Boudreault et al. (2015). The terrain height is classified as the median elevation $z_m$ of all ground or points, inside the $20 \times 20$ m$^2$ grid points, depending on if the grid cell was determined to be land or water. The forest height $h$ was estimated as $h = \max(z_i - z_m)$, where $h$ is the forest height and $z_i$ indicates the vertical coordinate of all points $i$ inside the grid point.

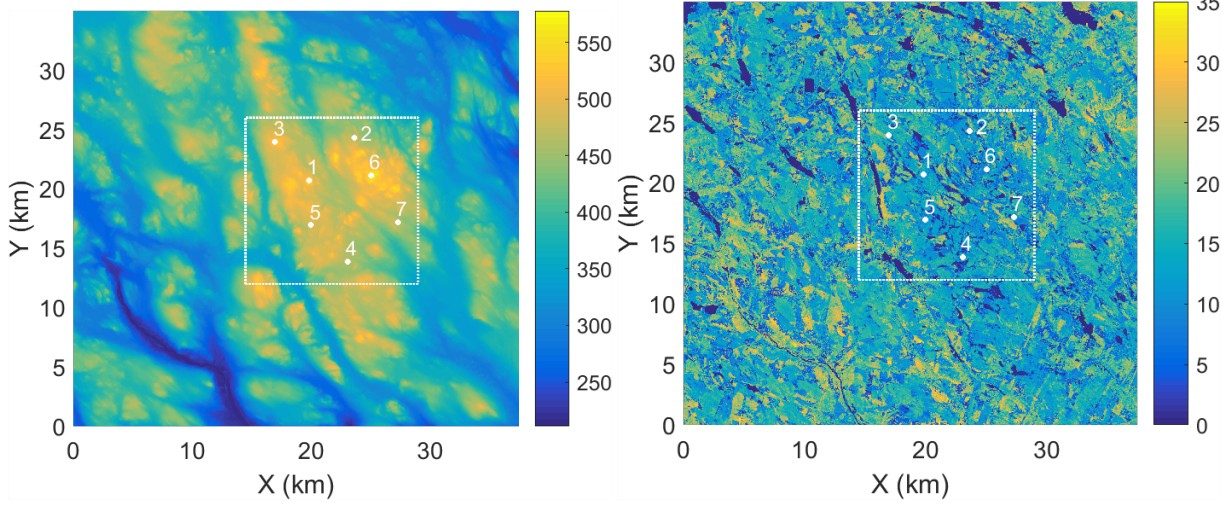

**Figure 1.** Terrain (left) and tree height (right) maps of the site derived from Airborne Laser Scans with a 20m resolution. The dashed rectangle indicates the area covered in Fig 2, and the numbered points show the locations of the seven masts.

Figure 1 shows the tree heights and elevation derived from the ALS processing, and an overview of the location of the seven meteorological masts used in this study. From the tree height map, it can be seen that the area is predominately forest, with $h$ ranging from 5 to 35 meters, but also contains areas of clearings. This makes the site ideal for testing the different as the





meteorological masts are impacted by both changes in tree height and forest edges. The focus of this study is on the forest parameters, but the terrain height will also change the forest's impact on the wind conditions, affecting the results. Therefore, the significant terrain variability of this site was desirable.

The baseline $20 \times 20$ m tree height map was based on the maximum tree height. Since the lidar beams do not necessarily hit the tree top, this tree height estimate underestimates the maximum tree height. A relationship between scanning density and maximum tree height was shown in Boudreault et al. (2015). For the site considered here, the scanning density was between 0.25 and 1 points per m$^2$, leading to an underestimation of the tree height of up to 2 m. Nilsson et al. (2017) created maps of the *mean* tree height from Airborne Laser Scans using the the 95 percentile of points located above 1.5m above ground, leading to a somewhat lower tree height. Compared to the uncertainties involved in the determination of roughness for the case when no tree height information is available, we consider this uncertainty to be of little importance.

### 3.2.2 An objective approach to estimate the roughness length and displacement height

The relatively simple conversion recommended by Garrat (1992) of $z_0 = 0.1h$ was used for converting directly between forest height and roughness length. To simplify the maps and reduce the number of roughness changes, for compatibility with WAsP, the forest height map was rounded to the nearest 5 m height $H$. Non-forested areas were identified as regions with vegetation heights lower than 2.5 m, and had their roughness was set to 0.1 m, while water areas were prescribed a roughness value of 0.1 mm. In summary, the roughness was parametrized as

$$z_0 = \begin{cases} 0.1H & \text{for } h \geq 2.5 \text{ m} \\ 0.1 & \text{for } h < 2.5 \text{ m} \\ 0.0001 & \text{for water areas.} \end{cases} \tag{2}$$

A sensitivity experiment was carried out using conversions of $z_0 = 0.2H$ and $z_0 = 0.05H$ to investigate how the roughness to tree height fraction impacts the modeling error. In addition to the calculation of roughness length, the forest height was used to calculate a zero-plane displacement height $d$, which was added to the terrain height for a subset of the WAsP modeling runs. The relationship between these parameters used in this study again came from Garrat (1992):

$$d = 2/3H. \tag{3}$$

The approach described above is called the Objective Roughness Approach (ORA).

### 3.2.3 Changing the map resolution

To investigate the impact of the map resolution and provide a fair comparison with the satellite-based maps, the $20 \times 20$ m resolution tree height maps were downgraded to 100, 300, 500, and 1000 m resolution roughness and displacement height maps by the following steps:

1. The forest height was re-calculated as the arithmetic mean of the forest height pixels in the higher resolution map.





2. Water areas were kept if the coarser resolution pixel consisted of more than 50 % of high-resolution water pixels.

3. The new forest heights were converted to roughness length and displacement height as in Eq. 2 and Eq. 3.

4. The resulting displacement height was linearly interpolated to a $60 \times 60$ m resolution terrain height map. However, in order to simulate a $20 \times 20$ m resolution case, we linearly interpolated the $60 \times 60$ m resolution terrain height map to a $20 \times 20$ m resolution before adding the displacement height map.

All resulting maps from these steps were exported as a Golden Software ASCII grid (.GRD) file for compatability with the WAsP Map Editor. Throughout the rest of the paper, the model simulations using these maps will be labelled as ORA followed by the resolution in meters.

For the simulations where the displacement heights were added to the terrain height, the anemometer heights (Table 1) need to be adjusted. This is done by subtracting the mast displacement height from the anemometer height. The mast displacement height was calculated the same way as it is in WindPRO, the arithmetic mean of all grid points within a 50 m radius of each mast. Table 4 shows the mast displacement heights for each ORA resolution, ORA maps that use a displacement height are labelled with an additional 'D'. All masts are located in small clearings, leading to generally low mast displacement heights, but at coarser resolutions the clearings are less resolved.

**Table 4.** Table with the displacement heights in meters determined from pixels within a 50 m radius around each mast location for each of the ORA maps.

| Abbreviation | Mast 1 | Mast 2 | Mast 3 | Mast 4 | Mast 5 | Mast 6 | Mast 7 |
|---|---|---|---|---|---|---|---|
| ORA20D | 2.46 | 5.08 | 6.39 | 5.09 | 6.67 | 4.77 | 6.40 |
| ORA100D | 5.98 | 5.43 | 7.87 | 8.55 | 8.13 | 6.30 | 8.53 |
| ORA500D | 8.11 | 7.41 | 10.09 | 7.57 | 7.59 | 7.01 | 8.34 |
| ORA1000D | 7.91 | 7.98 | 9.89 | 7.11 | 7.38 | 7.05 | 9.27 |

### 3.2.4 Conversion of raster maps to vector maps for WAsP

The WAsP model currently only accepts vector based roughness and elevation maps as input. Therefore, the raster based maps produced by the process described above need to be vectorized. This process was carried out using Version 11.20.5, Release D of the WAsP Map Editor for all ORA resolutions coarser than 20m. The Map Editor allows for the importing of raster based maps, which it then either contours, for elevation data, or converts to roughness change lines. For elevation data, raster data with a 60 m resoultion was converted to 10 m equidistant contours. For the roughness change lines, the option to keep the same roughness classes as in the raster file was used. To create the roughness change map, the Map Editor outlines all pixels with roughness change lines that represent the inside and outside roughness value, skipping those where both sides of the line would be the same. After processing the data, they were examined via a geographical information system (GIS) to ensure that the vector data provided a reasonable representation of the elevation and roughness raster data. For converting the 20m resolution





maps, the Terrain workshop, Version 1.1 Release D was used, because the Map Editor was not able to efficiently process the higher resolution data.

## 4   WAsP setup

WAsP version 11.6 was used in this study to simulate the wind resource at the different masts. In WAsP, the effect of changes in

roughness length on the wind speed around the site are modeled using the internal boundary layer zooming-grid (IBZ) model (Troen and Petersen, 1989; Sempreviva et al., 1990), while the speed-up effects from terrain elevation are simulated using the spectral model described in Troen and De Baas (1987); Troen and Petersen (1989). As input, WAsP requires an observed wind climate and vector maps of elevation and roughness.

WAsP uses the following modelling chain to simulate the wind climate at a one position, horizontal and vertical, from an

observed wind climate at another position. This modelling chain is extensively described in (Troen and Petersen, 1989), but here we briefly repeat the steps:

1. Fit a Weibull distribution to the observed wind climate for each sector, preserving the third moment of wind speed and the probability to exceed the mean wind speed.

2. Calculate the background wind profile using the geostrophic roughness and the observed wind profile, from which the

effects of the roughness changes, terrain height and atmospheric stability at the observational site have been removed.

3. Calculate the geostrophic wind speed (generalized wind climate) for predefined heights and roughness lengths using the geostrophic drag law in combination with the background wind profile.

4. Apply the effects of geostrophic roughness, terrain height, roughness changes and stability using this generalized wind climate for the predicted site.

The predicted wind climate can be quite sensitive to the choice of the standard heights and roughness lengths used in step 3. For this study, the predefined heights were here set to 3, 10, 30, 60, and 120 m above the surface, and the roughness lengths, were set to 0.0, 0.1, 0.4, 1.0, 3.0 m, which covers all possible roughness length values that occur in the maps. In the WAsP model, large water bodies are required to have a roughness length of 0.0 m, which are then internally converted to a value of $z_0 = 0.0002$.

The impact of varying topography, roughness changes and orography, is modeled as correction factors that are applied to the scale parameter ($A$) of the Weibull distribution of the wind climate in steps 1 and 4 of the model chain. For computational efficiency, WAsP filters the roughness changes to include only those that have the most significant impact on the wind profile. This is done, creating a distance weighted roughness length ($ln\ z_{0w}$), which is found by multiplying log-transformed roughness values with the exponentially-weighted distance from the mast to the roughness changes $x_k$ as,

$$\hat{x_k} = x_d \left[ 1 - \exp\left( \frac{-x_k}{x_d} \right) \right] \tag{4}$$




where $x_d$ is a decay length, which currently is set to 10 km. Then the most important $n$ nodes are found by fitting a step function that stops when either the maximum number of allowed steps $n_{max}$ is reached or the residual variance $RMS_{max}$ is below a specified threshold. The default values for $n_{max}$ and $RMS_{max}$ are 10 and 0.3, respectively. From this filtered set of roughness change lines, the internal boundary layer equations described in Sempreviva et al. (1990); Floors et al. (2011) are

applied to the reduced arrays to compute a speed-up factor in each sector.

To calculate the geostrophic wind, WAsP uses a corrected logarithmic wind profile, where the correction factors for internal boundary layer and orographic effects have been removed, in combination with the geostrophic drag law. The roughness length that is used in these relations is the so-called geostrophic roughness length $z_{0G}$, which is computed sector-wise from the mean of $\ln(z_0)$ and with a distance weight similar to Eq. 4:

$$\ln z_{0G} = \ln(z_{01}) \left[ 1 - \exp\left( \frac{-x_1}{x_d} \right) \right] + \sum_{k=2}^{N-1} \ln(z_{0k}) \left[ \exp\left( \frac{-x_{k-1}}{x_d} \right) - \exp\left( \frac{-x_k}{x_d} \right) \right] + \ln(z_{0N}) \left[ \exp\left( \frac{-x_N}{x_d} \right) \right], \quad (5)$$

where $z_{01}$ is the roughness length at the mast, $k = 2, ..., N-1$ denotes the $k^{th}$ roughness change from the mast location and $N$ is the last roughness change on the map.

In addition to the standard inputs, WAsP uses the long-term distribution of heat fluxes at the site to model the effect of atmospheric stability. Since no heat flux observations were taken at the site, the default heat flux distribution, with mean and

the standard deviation of $-40$ and $100\,\mathrm{W\,m^{-2}}$, respectively, was used over land. Over water, these values were set to $-8$ and $30\,\mathrm{W\,m^{-2}}$.

The performance of the different topographic maps was evaluated using cross predictions. A cross prediction is defined as the prediction of the flow from one observed wind climate, a specific mast and height, to another observed position, either another height on the same mast or an observed height on another mast. Cross predictions were made from all four heights at

six of the seven masts, at mast 6 the 80.7 m height was excluded due to the limited amount of data with QC = 1. After excluding self-predictions, i.e. predictions and inputs at the same mast and height, there were a total of 702 different combinations. The relative errors for each cross prediction were computed as a percentage from the observed ($obs$) and modelled data ($mod$) as $\delta = 100(mod - obs)/obs$ for both wind speed ($\delta U$) and power density ($\delta P$). It is important to include power density in the evaluation, since the production of wind turbines is determined by the available power. The total power density is calculated

by summation of the frequency weighted third moment of the Weibull distribution from each sector of the total of number of sectors $D$,

$$P = \sum_{l=1}^{D} 0.5 \rho f_l A_l^3 \Gamma(1 + 3/k_l), \quad (6)$$

where $\rho$ is a reference air density (here $1.225\,\mathrm{kg\,m^{-3}}$), $f$ is the frequency of occurence and $k$ is the shape parameter of the Weibull distribution. Kelly et al. (2014) showed that errors in $k$ can result in large errors in $P$.





## 5 Results

The results are presented in three sub-sections. First, the ORA roughness lengths will be compared with the roughness lengths
from the land-use based datasets. Second, the results of the wind data filtering algorithms will be shown. Finally, the results
from the WAsP cross predictions are shown.

### 5.1 Roughness maps

Figure 2 shows the roughness lengths of the four different land-use based datasets and two resolutions of the ORA data. It
can be seen that the small lakes in the eastern part of the domain are not well represented at resolutions of 300 m or more. In
addition to the different roughness length values, the forest edges and clearings are positioned differently across the different
datasets.

In the ORA maps the average roughness of the map varies between 1.42 and 1.46 m. The logarithmic average roughness
value over these map changed from 1.02 m ($20 \times 20$ m) to 1.39 m ($1000 \times 1000$ m) resolution. The impact of resolution is
discussed further in Sec 6.

It is clear from Fig 2 that the roughness lengths from the ORA maps are larger than from the land-use based maps. Addi-
tionally, the ORA data, in part due to the higher roughness values, has significantly more roughness changes. For example,
CORINE only has two forest roughness lengths, 0.4 and 0.5 m, while the ORA data represents forest roughness lengths in 6
different bins from 0.5 to 3.0 m.

These features can be seen more clearly in the histogram of the roughness lengths (Fig. 3), in this figure the ORA data has
the highest and lowest resolutions tested. The GLCC data set is dominated by a single land use class with $z_0 = 0.5$ m. The
MODIS data have slightly more detail and grid cells with $z_0 = 0.3$ m are most frequent. The CORINE data has the highest
resolution of the land-use based products, and has most grid cells with $z_0 = 0.5$. Note that the CORINE and ORA20 maps
include more variation of the roughness, which causes the geometric mean to be smaller than the arithmetic mean.

### 5.2 Wind distributions and profiles

To demonstrate the impact of the quality control that was described in Sec. 2, three distributions of wind speeds at different
stages of the quality control process are shown for the cup anemometer located at mast 1, 59 m ((Fig. 4). Due to the confiden-
tiality of the data, a normalized wind speed $\hat{U}$ was computed by dividing each 10-minute measurement by the mean wind speed
of the filtered data. The distribution before any filtering (Fig. 4a) shows a high frequency of measurements with very low wind
speeds, $< 0.2$, as well as an increase in wind speed occurrences between 0.2 and 1.0, when compared to the distribution in Fig.
4b that includes only data flagged with QC = 1. These differences likely reflect the icing of the cup anemometers. Because of
the large number of near-zero wind speeds, the Weibull distribution from the WAsP method in Fig. 4a gives a rather poor fit.
The fit was improved for the distributions including only data with QC = 1 (Fig. 4b), which includes 59% of the original data
set, and requiring that QC=1 for all cup anemometers simultaneously (Fig. 4c), which retains only 16 % of the original dataset.
The mean wind speed for the three distributions were 0.85, 0.97 and 1, highlighting that although there was a large change in




**Figure 2.** Roughness maps, colored by roughness value. Open circles show mast locations.

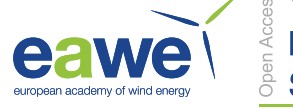
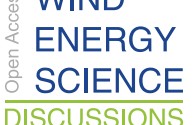


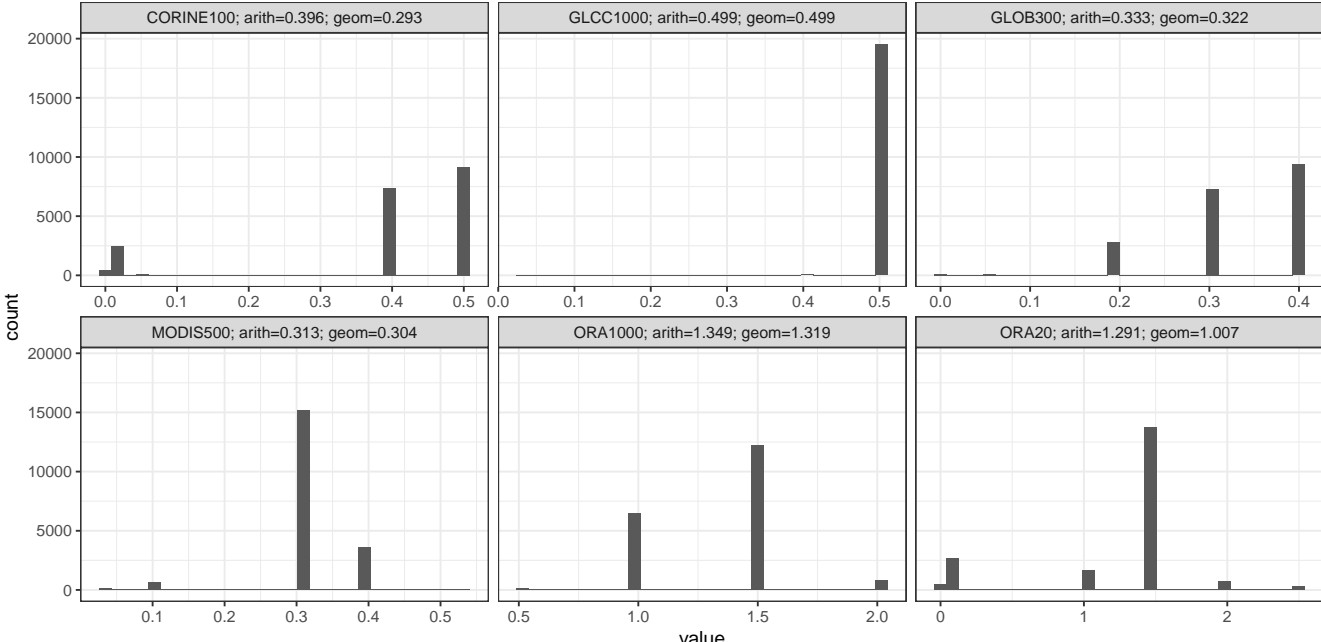

**Figure 3.** Histogram of roughness for different datasets over the same domain as shown in Fig.2. The header of each graph also contains the arithmetic and geometric means of roughness for the region.

dataset size when requiring QC = 1 for all cups simultaneously, there was a small change in mean wind speed compared to QC = 1 for a single location. This relationship was found for all masts and heights. All 27 anemometers showed an increase between 1 and 6 % when demanding that QC = 1 for all cup anemometers simultaneously, which shows that the wind climate in the final dataset used for the validation is representative for the site.

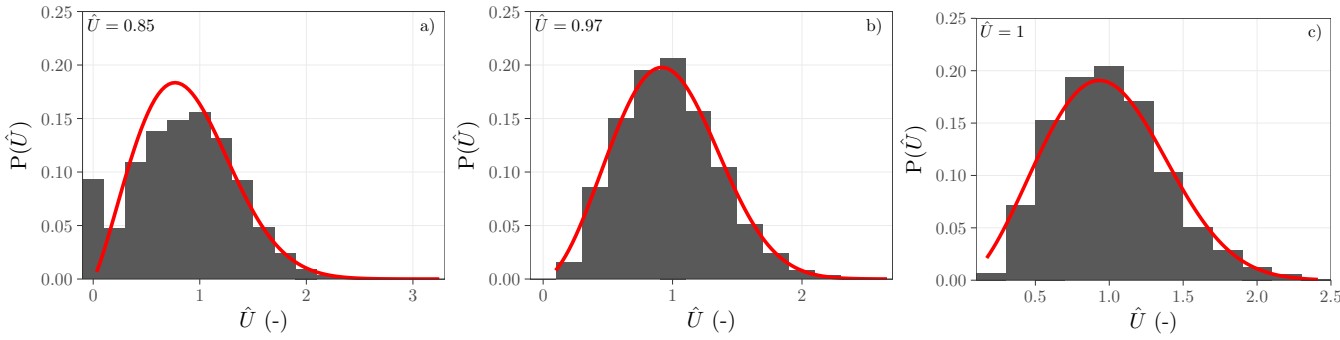

**Figure 4.** Histogram of the observed all-sector normalized wind speed $\hat{U}$ (see text) at mast 1 at 59 m agl without any filtering (a), after selecting data with QC = 1 with 59 % of the data (b) and during the times when all masts simultaneous had QC = 1 with 16 % of the data (c). The red line denotes the Weibull distribution that the WAsP fits for this histogram.





**Figure 5.** The all-sector modelled mean wind profile (lines) and the observations (points) at the seven masts. All the profiles were obtained by using the observed wind climate from mast 1 at 59.0 m and have been normalized by the mean wind speed observed there.

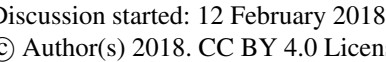



Fig. 5 shows the vertical profiles of mean wind speed for all anemometer positions using the observed wind climate from mast 1 at 59 m. The data has been normalized with the mean wind speed from the observed position. In addition to the land-use based simulations, the ORA runs include the highest resolution maps both with displacement heights included (ORA20D) and without (ORA20). The impact of introducing a displacement height can be observed near the canopy, where the wind speed

is much lower than when a displacement height is not included. At masts 1, 3, and 7 the ORA20D simulations are closer to the observations than without displacement, while they are comparable at the other masts. At higher heights, the differences in mean wind speed between different tests are smaller. The worst predictions occur at mast 6, however, this site is at a higher hill than the other masts, so this difference could be due to orographic speed-ups.

### 5.3    WAsP sensitivity to different roughness maps

The first test performed was to investigate the sensitivity of the ORA method to different tree height to roughness length conversions in Eq. 2. In this sensitivity test, simulations were performed using maps where $z_0 = H/5$, $z_0 = H/10$ and $z_0 = H/20$ with and without applying a displacement length. The resulting mean absolute error, $\overline{|\delta U|}$, of the 702 cross-predictions are shown in Fig. 6, for these tests as well as a sensitivity test of the CORINE land-use based roughness.

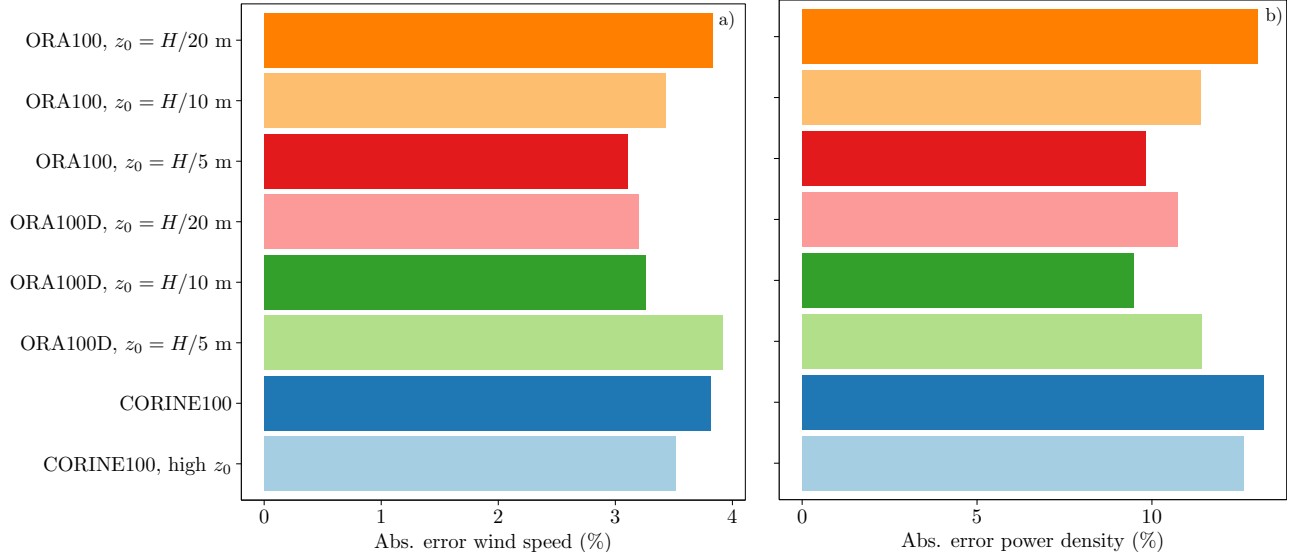

**Figure 6.** a) The mean absolute error in wind speed (%) or b) power density for all simulations using a landuse description with a resolution of 100 m

For the roughness maps without a displacement height, it can be seen that increased roughness lengths result in lower errors,

i.e. $H/20$ has the highest error and $H/5$, which corresponds to a mean roughness length of more than 2 m, has the lowest. When a displacement height is applied, better representing the location, $H/5$ test has the largest errors, while the tests using $H/10$ and $H/20$ maps have very similar $\overline{|\delta U|}$. When looking at power density, it was found that that the ORA100D simulations with



$H/10$ and $H/20$ had a $\overline{|\delta P|}$ of 9.48 % and 10.74 %, respectively. This study shows that $z_0 = H/10$ is a reasonable modelling choice.

After finding that higher roughness lengths lead to better results, and noting that the roughness maps based on land use classes (GLCC1000, MODIS500, CORINE100 and GLOB300) are characterized by much lower roughness lengths than the
ORA maps (Fig. 3), one could hypothesize that increasing $z_0$ in the satellite based products to a more realistic value would improve the results. The CORINE100 data was chosen for this since it could be seen that the CORINE100 map had a realistic representation of the land use around the site (Fig. 2).

To test this, the roughness length of the forest classes (i.e. areas with $z_0 = 0.4$ m and $z_0 = 0.5$ m) in the CORINE100 map were increased by a factor of three to 1.2 and 1.5 m, respectively, which is approximately the roughness length observed in
maps using the ORA approach (see Fig.3). The simulation using this roughness map is labelled with "high $z_0$" and had an error of $\overline{|\delta U|} = 3.5$ %, which is closer to the ORA100 test's error of $\overline{|\delta U|} = 3.4$ % than the standard CORINE100 test with an error of $\overline{|\delta U|} = 3.8$ %. Therefore, the performance of the land-use based maps could potentially be improved for this site by increasing the roughness length. However, the factor of three was chosen based on knowledge of the roughness length from the ORA maps, so in the absence of tree height measurements it could be difficult to set the correct value.

The next sensitivity test was designed to investigate the effect of the data filtering on the roughness change model. This was achieved by changing the limit of $\mathrm{RMS_{max}}$ to $\approx 0$ and varying $n_{max}$ from 0 to 10, which causes the model to include only $n_{max}$ roughness changes after filtering. Note that using 0 roughness changes disables the internal boundary layer model, which means that all observed differences are due to a different $z_{0G}$ (see Sec. 4). To isolate the impact of taking more roughness changes into account, all simulations were normalized by the $\overline{|\delta U|}$ from simulations with 0 roughness changes.

Fig. 7 shows the normalized $\overline{|\delta U|}$ for different numbers of roughness changes. It can be seen that for most roughness maps, the model errors generally decrease from 0 to 2 $n_{max}$, and for larger $n_{max}$ the ratio is more constant. This indicates that in this case only two roughness changes contribute significantly to WAsP modelling at each mast. This is consistent with the best practice for hand digitizing roughness maps, which states that only the most important roughness changes need to be digitized (Mortensen, 2016). It should be noted that $n_{max}$ is likely site-specific and that using $n_{max} > 2$ could still give significant
improvements at other sites.

By comparing the different ORA resolutions, the impact of resolution on the roughness change model can be examined. It was found that the simulation using the ORA20 map had the largest reduction and $\overline{|\delta U|}/\overline{|\delta U_0|} \approx 0.86$, while the coarser resolution ORA1000 map did not have as large of a reduction in error ratio, $\overline{|\delta U|}/\overline{|\delta U_0|} \approx 0.94$ using $> 2$ roughness changes. This shows that the resolution is important for roughness change modelling, which was expected since the higher resolution
maps can better resolve the precise location of a roughness change. Interestingly, the GLOB300 simulations have $\overline{|\delta U|}/\overline{|\delta U_0|} > 1$, indicating that the map with $n_{max} = 0$ performs best. This means that the model results can lead to larger errors when including roughness changes that do not well represent the modeling area.

To further illustrate the effect of using different maps, we can plot the $\overline{|\delta U|}$ as a function of $\overline{z_{0G}}$ (Fig. 8), where $\overline{z_{0G}}$ is the all sector geostropic roughness length, which was calculated as the geometric mean of $z_{0G}$ from each sector. Since there are
no displacement heights available for the land-use based maps, the displacement height was not used in this comparison. As



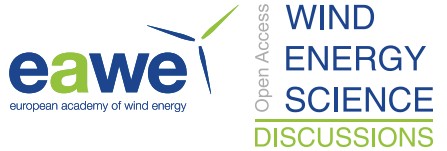

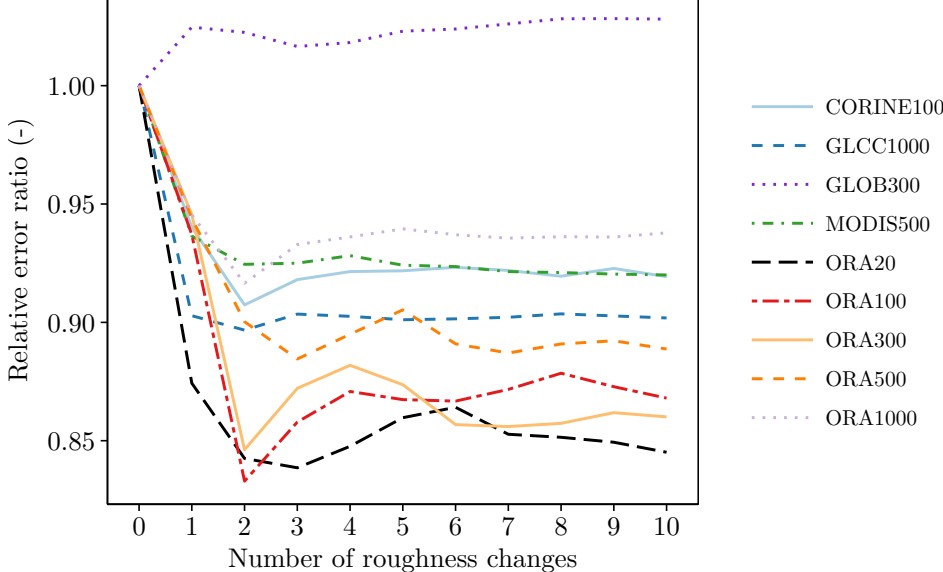

**Figure 7.** The mean absolute error in wind speed, $\overline{|\delta U|}$, as a function of the number of significant roughness changes $n$, where for each $n$ we normalized by the mean absolute error in wind speed from simulations with the roughness change model switched off.

expected, the satellite based products have a much lower $\overline{z_{0G}}$ than the ORA maps. There is also a relationship between the model error and $\overline{z_{0G}}$, in that the MODIS500 map has the highest $\overline{|\delta U|}$ of $\approx 4.4$ %, with the lowest $\overline{z_{0G}}$, whereas the ORA1000 map has the highest $\overline{z_{0G}}$ and the lowest $\overline{|\delta U|}$ of $\approx 3.4$ %.

The effect of including a displacement height for the ORA maps at different resolutions is shown in Fig 9. The roughness
change model is used with the default settings (see Sec. 4), so these results are impacted by both the roughness changes and the different values of $z_{0G}$. The ORA20D and ORA100D have a $\overline{|\delta U|}$ that is $\approx 0.2\%$ lower than the ORA20 and ORA100, respectively, so they benefit from including a displacement height.

At coarser resolutions, adding the displacement height increases $\overline{|\delta U|}$ by $\approx 0.5$ %. he displacement height at all masts using the ORA1000 map is significantly higher than for the ORA20 map (Table 4). This suggests that the displacement height for the
coarser resolutions is likely too high, because the forest clearing around the mast is not resolved. Therefore, it is recommended that the displacement height correction should only be performed when the resolution of the map is sufficient to represent the effect of forest clearings near the mast.

### 5.4   Summary of using different roughness maps

In this section, the overall performance of WAsP simulations using satellite based maps and the proposed ORA approach is
summarized. Thus far, only $\overline{|\delta U|}$ was used for comparing the model results, since the comparisons were largely investigations of the model behavior. However, for an annual energy production estimation, it is more relevant to quantify map-related errors





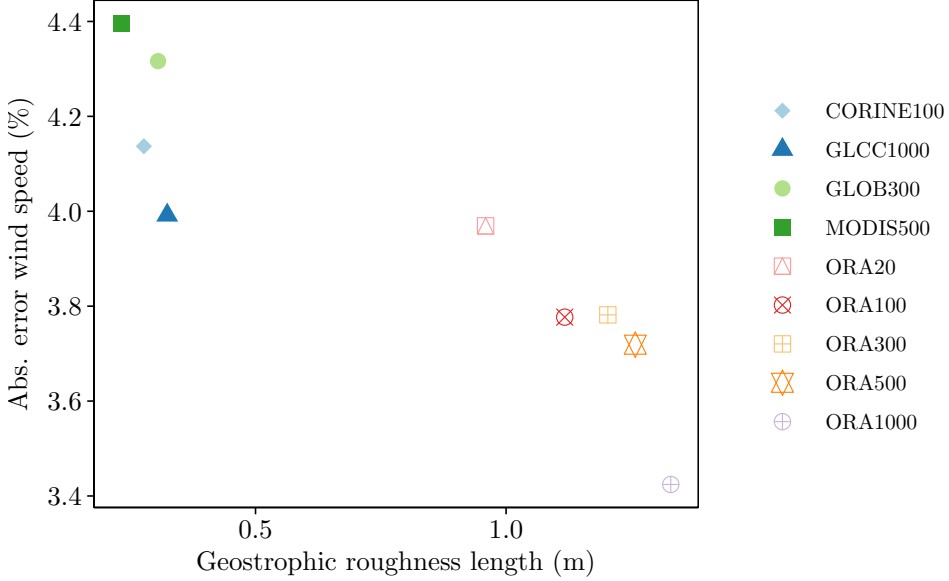

**Figure 8.** The mean absolute error in wind speed as a function of the geostrophic roughness length when using the default WAsP settings but with the roughness change model switched off.

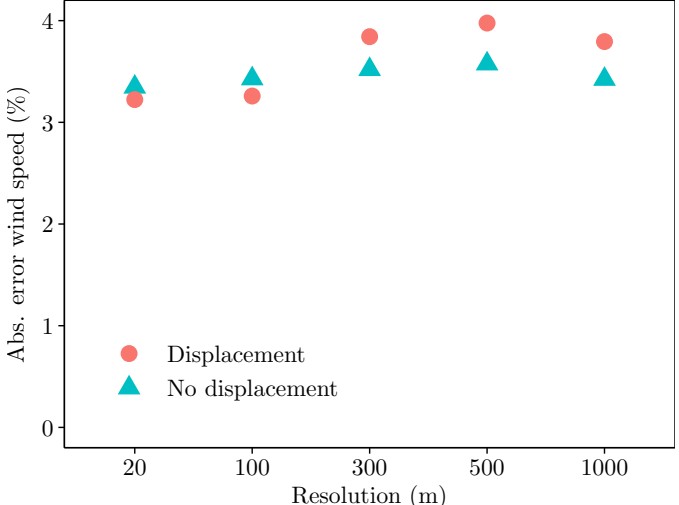

**Figure 9.** The mean absolute error of the simulations with the ORA maps with and without applying a displacement height for different map resolutions.




on $\overline{|\delta P|}$ (Eq. 6) to assess the potential performance of the wind turbine. Therefore, to evaluate the overall performance of the modelling with different maps, both $\overline{|\delta U|}$ and $\overline{|\delta P|}$ will be used.

Fig. 10 shows the distribution of $|\delta P|$, for each of the cross-predictions, binned in three categories for all four land-use based maps and the highest resolution ORA map with a displacement height. For this analysis, the cross predictions are broken into

two types, horizontal predictions (Fig. 10a) are those where the observed wind climate and the predicted wind climate are from different masts, while the vertical cross-predictions are those where they come from the same mast (Fig. 10b).

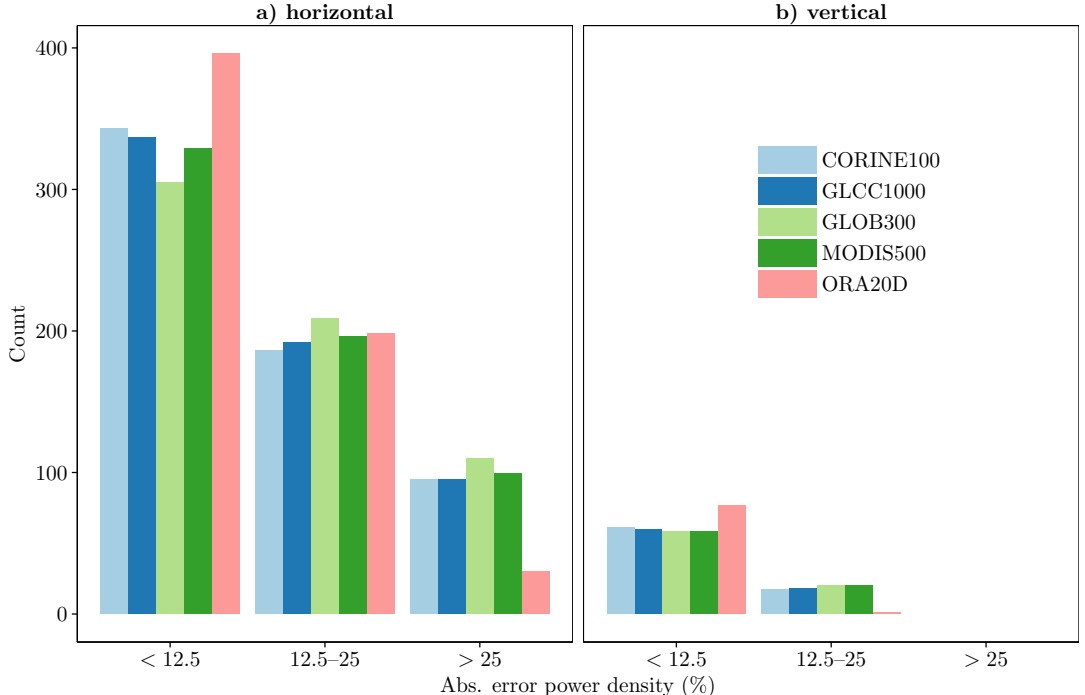

**Figure 10.** Absolute percent errors of the power density for 702 cross predictions using simulations with different maps.

Ideally, we want as many cross-predictions with $|\delta P| < 12.5\%$ as possible. For the horizontal cross-predictions, there are $\approx 20\%$ more predictions in this category for the ORA20D run than for the land-use based maps. On the other hand, the number of very high errors, $|\delta P| > 25\%$, is greatly decreased, for the horizontal cross-predictions, when using the ORA20D map, from

$\approx 100$ when using the CORINE100 and GLCC1000 map to $\approx 20$ when using the ORA20D map.

For the vertical cross-predictions (Fig. 10b), the errors are much smaller than for horizontal extrapolations because there are the same topographic errors for the input and the predicted anemometers. However, as in the horizontal cross-predictions, the ORA20D run has fewer high error predictions than land-use based roughness map; the ORA20D simulation had only 1 error $> 12.5\%$, whereas for all other land-use products they occurred at least 11 times.

Even when comparing maps with the same resolution (not shown), i.e. the CORINE100 with the ORA100 simulation, the number of cross-predictions with errors larger than 25% decreases from 75 to 44, a 41 % decrease. Using the ORA300





compared to the GLOB300 simulation decreases these errors by 51 % and using the ORA500 compared to the MODIS500 decreased them by 40 %.

Table 5 provides summary error statistics of the results from the 702 cross predictions for all land-use, ORA and ORA with displacement simulations. In addition to the mean absolute error of $\delta U$ and $\delta P$, which has been shown previously, the mean

bias and root mean square error (RMSE) have been included. The RMSE penalizes high errors more than the mean absolute error, and therefore identifies simulations that have a narrow error distribution.

**Table 5.** Summary of the error metrics using all model runs (702 cross predictions). Defining $y$ as modelled and $x$ as observed variable with a line denoting a mean, the mean absolute error is defined as $100\overline{|(y_i - x_i)/x_i|}$, the mean bias as $100\overline{(y_i - x_i)/x_i}$ and the root-mean-square error (RMSE) in percent of the wind speed $U$ and the power $P$ as $\sqrt{\overline{\left(100(y_i - x_i)/x_i\right)^2}}$. The lowest value of an error metric is denoted in bold.

|            | Mean bias $U$ (%) | Mean abs. error $U$ (%) | Mean abs. error $P$ (%) | RMS error $U$ (%) | RMS error $P$ (%) |
|------------|-------------------|-------------------------|-------------------------|-------------------|-------------------|
| CORINE100  | 0.30              | 3.82                    | 13.19                   | 5.02              | 17.09             |
| GLOB300    | 0.31              | 4.27                    | 14.41                   | 5.48              | 18.49             |
| GLCC1000   | 0.51              | 3.61                    | 13.14                   | 4.91              | 17.05             |
| MODIS500   | 0.22              | 4.10                    | 13.60                   | 5.21              | 17.35             |
| ORA20      | 0.50              | 3.35                    | 11.15                   | 4.27              | 14.10             |
| ORA100     | 0.34              | 3.43                    | 11.39                   | 4.36              | 14.24             |
| ORA500     | 0.17              | 3.58                    | 11.57                   | 4.51              | 14.39             |
| ORA1000    | **0.06**          | 3.42                    | 10.61                   | 4.27              | 13.16             |
| ORA20D     | 0.49              | **3.22**                | 10.03                   | 4.07              | 12.40             |
| ORA100D    | 0.36              | 3.26                    | **9.48**                | **4.03**          | **11.70**         |
| ORA500D    | 0.23              | 3.98                    | 11.67                   | 4.98              | 14.48             |
| ORA1000D   | 0.10              | 3.79                    | 11.12                   | 4.72              | 13.73             |

All ORA based simulations have lower $\overline{|\delta P|}$ than any of the simulations using land-use based maps. The same is true for the RMSE of $\delta P$, where the improvement of the simulations with the ORA roughness maps is even larger, due to the reduced number of large errors as seen in Fig. 10.

Furthermore, it was found that the ORA100D simulation has the lowest $\overline{|\delta P|}$, RMS of $\delta U$, and RMS of $\delta P$, whereas the ORA20D has the lowest errors in $|\delta U|$. The ORA20D map results in lower errors due to a more accurate roughness change description (Fig. 7), and due to the application of a displacement height (Fig. 9). However, the $z_{0G}$ was lower than that used for better performing maps in the no-roughness change tests (Fig. 8). From these results it's clear that the ORA maps provide better results, both due to the higher $z_{0G}$ and because of the increased detail of the forest structure. For this site, a best-performing

map could likely be created by using the 20 m resolution map with a slightly higher $z_0$.

The land-use based land cover simulations perform similarly despite different resolutions and vastly different descriptions of the land use. For example, the GLOB300 and CORINE100 simulation both have a rather high RMSE for both $U$ and $P$, despite having a high resolution. This highlights that resolution is not a panacea, but only improves performance when the




surface is accurately represented. For example, the GLCC1000 simulation has almost no roughness changes (see Fig.2), but has the lowest RMSE in $U$ and $P$ out of the satellite based products. Also we noted that the CORINE100 with a high $z_0$ (see Sec. 5.3) reduced $\overline{|\delta P|}$ to 12.63 % and the RMSE of $\delta P$ to 15.76 % (not shown), however, both are still significantly higher than all of the ORA runs. This shows that tweaking $z_0$ of certain land use classes in existing satellite-based maps will only
partially reduce the model errors.

In Table 5, the mean bias of $U$ is very close to zero for all model runs. The ORA1000 simulation has the lowest mean bias of $U$. One would expect a difference between runs with a very low and very high roughness, for example the CORINE100 and ORA100 run. However, it can be seen that for both of these runs, the bias is close to zero. This is likely because the cross-prediction includes both upwards and downwards predictions. This results in errors that will cancel each other out; e.g. a too
low prescribed roughness results in an over-prediction for an upward extrapolation, but an under-prediction for a downward extrapolation.

## 6   Discussion

The presented results for the mean prediction error in Table 5 show a complex inter-dependency of roughness magnitude (Fig. 6 and Fig. 8), imprecision of roughness lines due to either limited map resolution or a coarse land-use classification (Fig. 7),
and inclusion (or not) of displacement height (Fig 5 and Fig.9). Nevertheless, the ORA approach showed consistently improved results relative to the roughness maps based on land-use classification for the studied site. This demonstrated robustness of the ORA approach should make it attractive to use for turbine siting. Given the relatively small processing time needed to make the maps, and the fact that all maps could be loaded into WAsP and predictions finalized within a few minutes, the approach is promising. At this site, it turned out to be better to use a land-use map with hardly any roughness change lines than to use
products with more detailed, but incorrect information, GLCC1000 versus GLOB300, when comparing only the commercially availible maps (Table 5).

### 6.1   Ideas for further improvements

As argued by Jackson (1981) and Raupach (1994), the forest density in addition to the forest height is likely to influence the optimal ratios of $z_0/H$ and $d/H$. Since forest density in terms of Plant Area Index $PAI$ can also be estimated from
the airborne lidar data (Boudreault et al., 2015), it would be interesting to investigate a more refined roughness classification based on this parameter. It was also demonstrated that raising the roughness value in the CORINE data reduced the mean prediction error compared to the original roughness conversion (Fig. 6), indicating that this potential "quick fix" could result in significantly reduced errors. Without access to forest height information, it will however be hard to justify the exact level to which the roughness should be adjusted to. Since imprecise roughness change lines can lead to an increase in prediction
error, as seen when using GLOB300 data (Fig. 7), a simpler way of reducing the error could be to smooth the map and increase the geostrophic roughness in order to achieve lower error, as demonstrated for the ORA maps in Fig. 8. The results in Fig. 8




reinforce the recommendation to use high values of roughness length for forested areas, even for landscapes where all mast observations are affected by the forest.

When downgrading the resolution of the ORA maps, the arithmetic average of the roughness remained nearly-identical, because the starting point for all the ORA maps was the tree height, whereas the presence of clearings in the high-resolution

ORA maps reduced the geostrophic roughness (Fig. 8), since the WAsP method uses the geometric average (see *e.g.* Taylor, 1987, for a motivation and more background). In forested areas, the use of geometric averaging could be particularly problematic, since, in reality, the presence of clearings, which have low roughness lengths, tend to increase the overall turbulence levels, and thereby, increase the aggregated roughness of the landscape (Lopes et al., 2015). It should also be noted that the log-space averaging approach has limitations for low vegetation, since it omits the significant non-equilibrium effects of roughness

aggregation (Hasager and Jensen, 1999).

Because of the log-averaged aggregated geostrophic roughness, it would be of interest to systematically investigate the roughness value chosen for the clearings. The value of $z_0 = 0.1$ m, used in this study, was assessed to be low for a newly cleared forest area, but reasonable-to-high for low-vegetation wetland. If a larger roughness length was used for the clearings, it would be expected that $z_{0G}$ would also be increased, which would reduce the error associated with the geostrophic roughness

closer to the level of the ORA1000 map. There is, however, a risk that increasing the value could affect the filtering of significant roughness changes, which could thereby increase the related error.

## 6.2   Does a better map make a significant difference?

The reduction in *mean* prediction error for both the mean wind speed and power density is only a few percent. However, as demonstrated in Fig. 10, this reduction reflects changes over a wide distribution of cross-prediction errors. We see the

demonstrated reduction of the large errors as one of the main advantages of the ORA method, since such large errors in the predicted wind climate could result in significant costs when estimating the performance of a wind turbine throughout its lifetime. Whereas large errors could result in higher lifetime costs of turbine operation at any site, Enevoldsen (2017) revealed that wind turbines located in forested areas are more prone to experience fatigue loads, vibration errors, and lower annual energy productions than other sites. Therefore, an improved wind climate prediction could be of high consequence for the

forested landscape.

With this in mind, it is recommended that effort is put into finding accurate and updated information on the forest height and structure, when evaluating an optimal location for wind energy exploitation in forested areas to minimize the risks inherent in project evaluation. For the reasons stated above, we also recommend investigating the sensitivity of the observed wind climate to the relationship between $z_0$ and $H$, since the ratio used here may not be optimal for other forests.

## 6.3   Availability of tree height information data

To the authors' knowledge, databases of freely accessible ALS data already exists for Denmark, Finland, and England. In other countries, such as Sweden, data are accessible at reduced cost (compared to a new campaign) for international researchers and companies, whereas they are freely accessible for national researchers.



The processing of lidar data to tree heights is relatively simple, and can be performed on a normal desktop computer, for small areas. In addition, future drone-mounted lidars and satellite-derived products (Stone et al., 2016) have the potential to provide high-accuracy tree height or land cover information for relatively low cost. This will enable better access to such data, and provide for the ability to assess changes over time.

## 7 Conclusions

This study has demonstrated a new way of introducing high-quality roughness maps derived from ALS scans of forest areas into the WAsP and WindPRO software. The steps involved can be reproduced by siting engineers with access to a reasonably powerful computer. By examining wind conditions at seven meteorological masts in a forested area in Sweden, it can be concluded that simulations using the tree height-based roughness maps resulted in a closer agreement with observational data
when compared to those that used standard land-use based online sources. The lowest error was achieved when using the highest detail maps, 20 m and 100 m resolutions, and including the displacement height via the terrain height (Fig 9). A sensitivity test found that using $z_0 = H/10$ with a displacement height provided the best results for converting tree height to roughness length when the displacement was used, but the conversion of $z_0 = H/5$ performed better without a displacement height. In addition, while roughness length maps with a high resolution benefited significantly from applying a displacement height, it did not
improve the results when using maps with a coarser resolution than 100 m. The highest resolution ORA data provided the best results, but an improvement over the land-use based map simulations was also achieved when lowering the spatial resolution to 1000 meters. It can therefore be suggested that siting engineers and practitioners of wind resource assessment in forested areas should collect appropriate site data that describe the forest height, type, and density. It is furthermore recommended to apply the ORA conversion setup for heterogenous forests dominated by coniferous evergreens such as spruces and pines.

*Code and data availability.* The wind data used in this study are confidential and unfortunately not made public. The scripts that were used for making the figures and analyzing the data are available at github: https://github.com/RogierFloors/Rscripts

*Author contributions.* ED and JA processed the tree height and cup anemometer data. ED, PE and ND made the maps. RF performed the modelling and made the plots. RF, ED, ND, PE and JA wrote the paper.

*Competing interests.* Rogier Floors, Neil Davis and Ebba Dellwik work at DTU Wind Energy, where the WAsP is maintained, developed
and sold.





*Acknowledgements.* For financial support we thank the New European Wind Atlas project (The European Commission funded, FP7-ENERGY.2013.10.1.2) and the InnoWind project (Grant Number 6172-00004B, funded by the Danish Innovation Foundation. JA acknowledges StandUp for Wind, a part of the StandUp for Energy strategic research framework.).




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
