# Peer review of "From lidar scans to roughness maps for wind resource modeling in forested areas"

_Wind Energy Science, 2018_

## Referee Comment (RC1) · Anonymous Referee #1 · 1 Mar 2018

**From lidar scans to roughness maps for wind resource modeling in forested areas**

**by Floors et al.**

The paper presents different ways of taking surface roughness and tree height into account when using WASP.

I think it's an interesting paper and I recommend that is should be accepted.

Some detailed comments are given below.

p. 2, line 20   Reads *driven by lines of roughness change lines*. Is this correct?

p. 7, Table 3   What is the relation between *Equivalent Roughness Length (m)* and $z_0$ in Eq. 1?

p. 3, line 3, typo   It reads *height was used, therefore, the displacement*

p. 8, bottom, typo   It reads *testing the different as the*

p. 9, line 15, typo   It reads *had their roughness was set to 0.1 m*

p. 9, below eq. 2   It reads *the forest height was used to 20 calculate a zero-plane displacement height d*. I don't understand. Why it the displacement height related to the tree height?

---

## Referee Comment (RC2) · Anonymous Referee #2 · 18 Mar 2018

The manuscript does an excellent job at tackling the issues of data availability in land use maps for the calculation of wind resources in forested areas. The scientific significance and quality of the manuscript are both very high; I believe some sentences should be rewritten to be made clearer, and there are some minor inaccuracies with the use of English. I therefore recommend that after some minor technical corrections that manuscript be accepted.

Some specific comments below.

page 1, lines 10-13 (p1l10-13): While not unclear, sentence a bit convoluted and difficult to follow. Please rewrite.

p2l3: Please spell out WAsP and add website p2l13: substitute colon between references (Jackson (1981); Raupach (1994)) with 'and'. p2l24: In a later study, Dellwik et al. (2006) demonstrated...

p3l1: add website for WindPRO p3l7: ..impact of adding d to terrain data on model predictions. p3l18: ...about forest characteristics... p3l19: During the last decade, ALS... p3l26: ...based on standard land-use... p3l30: ...how WAsP handled... ...the speed of WAsP calculation. p3l34-35: replace'(Sec. 2) with 'Section 2 (i.e. no brackets, spell out Section). Also spell out Section in Section 3.

p4l5-8: Sentence not unclear but long and convoluted. Please rewrite. p4l24: Section 5.

p6l9:frequently p6l15: Why Carroll et al., 2010 while in Table 2 it's DiMiceli et al., (2011)? p6l7-16: Please add survey dates for the 4 land-use datasets. p6l17: Wind-PRO p6l18: Table 3 (i.e. capitalise 'Table') p6-Table 2: Suggested new title: 'Summary of the different land-use data sources used for creating the roughness maps' Capitalise all letters in CORINE; perhaps also add that it's European data Capitalise all letters in MODIS

p7-Table 3: Suggested new title: 'Land use category and prescribed values of roughness length for each land-use dataset' CORINE section. Last line: Broadleaved. What is complex cultivation under CORINE? Maybe add simple explanation to table. GLOB-COVER section. What is 'pen' mixed broadleaved? Global Land Cover section. First line: substitute 'needleleaf' with 'needleleaved'

p8l16: Figure 1 shows the tree heights (right panel) and elevation (left panel) derived... p8l18: This makes the site ideal for testing the difference... Please also state what difference. p8-Figure 1: Please use same ticks in axes X and Y. Also please specify that the units for elevation and tree heights are in m.

p9l15: ...and had their roughness set to 0.1m... p9l16: m (i.e. not mm) p9l24: 3.2.3 Changing map resolution p9l25: To investigate the impact of map resolution...

p10l6: ...file for compatibility with WAsP... p10l8: Maybe add example in brackets at end of sentence, e.g.: '(e.g. ORA20) ? p10l12-13: I would enclose in brackets the sentence: 'ORA maps that use...with and additional 'D'. p10l17: ...described above needed to... p10-Table 4. Suggested new title: Displacement height (m) determined from pixels...

p11l7: as for reference in page 2, substitute colon between references (Troen and De Baas (1987); Troen and Petersen (1989)) with 'and'. p11l10-11: 'This modelling chain is extensively described in Troen and Petersen (1989). Here we briefly summarise the steps:' p11l21: Please remove comma after roughness lengths p11l28: 'This is done by creating distance-weighted...

p12l1: Please briefly mention what the nodes are and what they represent. p12l4: as for references in page 2 and page 17, substitute colon between references (Sempreviva et al., (1990); Floors et al. (2011)) with 'and'. p12l20: '...six of the seven masts, but at mast 6...' p12-eq.6: I take A is the scale parameter of the Weibull distribution. Please make that explicit in the eq. description.

p13l12: Please spell out Section p13l17-18: Sentence not unclear but needs to be improved. Also please conjugate in plural those verbs with data as subject (e.g. data have, not data has) p13l23: Please spell out Section p13l24: too many brackets for Fig.4 p13l26: is filtering the QC-filtering? Maybe worth specifying? p13l27: Please put '< 0.2 in brackets, not between commas. p13l28: please remove 'flagged'

p14-Figure2. Suggested new title: Roughness maps from land-use datasets, and from ORA at 100m and 500m resolutions, colored by roughness values. Open circles...

p15. Figure 4: please remove 'the' before WAsP.

p17l12-13: Not unclear but should be made clearer. p17-Figure6. Suggested new title: a) Mean absolute error in wind speed (%) and b) power density...

p18l18: Please spell out Section p18l29: 'This shows that resolution is important...'

p19l8: 'The displacement height...' p19l16: '..investigations of model behavior'. p19-Figure 7. The title describes the graph as mean absolute error, while the Y axis on the graph reads 'Relative error'. Please correct accordingly.

p20-Figures 8 and 9: Please remove 'The' at the beginning of titles.

p21l5: '...into two types. Horizontal predictions ...' p21l16: remove space between 41 and %

p22l10: Should it be RMSE rather than just RMS? p22l17: '...different descriptions of land use.'

p23l20-21: '...products with more detailed, but incorrect information (e.g. GLCC1000 vs GLOB300) when comparing only the commercially available maps.' p23l23: 'As argued by Jackson (1981) and Raupach (1994), forest density and forest height are likely to influence the optimal...' p23l24-25: '...Plant Area Index (PAI) can also be estimated from airborne data...'

p24l4: '...ORA maps was tree height, whereas...' p24l7: '...problematic since - in reality - the presence of clearings... p24l8: '..the overall turbulence levels and thereby increase the...' p24l26-27: 'updated information on forest height and structure when evaluating...'

p25l18: '...appropriate site data that describe forest height, type, and density.' p25l19: I don't understand why spruces and pines are brought into the paper so abruptly (and at the very end of the manuscript!). Is it essential to talk about spruces and pines at this point? p25l24: '...DTU Wind Energy, where WAsP is maintained, developed, and sold.'

p26: Check positioning of brackets in Acknowledgements: there seems to be a mistake: bracket required after '(Grant Number 6172-00004B', not at the end of the sentence.

References: Is it correct that he publication year is at the end? Please check the

journal's suggested citation standard (I could not find it)

---

## Author Comment (AC1) · 24 Apr 2018

**1   Reponse Reviewer 1**

The paper presents different ways of taking surface roughness and tree height into account when using WASP. I think it's an interesting paper and I recommend that is should be accepted.

*We thank the reviewer for the comments and have made modifications accordingly (see below). The line numbers all refer to the track-changed manuscript where deletions are indicated in red and new text is indicated in blue.*

Some detailed comments are given below.

[Figure]

- p. 2, line 20 Reads driven by lines of roughness change lines. Is this correct?

We corrected this to: *"Heterogeneity in roughness is modelled through a roughness change model, which consists of an internal boundary layer model that is driven by lines of roughness changes on a map. The speed-up effects due to changes in terrain height are taken into account by a flow-over-terrain model."*

- p. 7, Table 3 What is the relation between Equivalent Roughness Length (m) and z0 in Eq. 1?

The word "equivalent" was confusing here, because the roughness length in this table are simply assigned to the map before applying it in the model chain as described in lines 9–19 on page 11. We therefore changed "Equivalent" to "Prescribed". Eq. 1 is not specifically used in WAsP and to be more explicit we added a reference to the vertical profile equations (page 11, line 10).

- p. 3, line 3, typo It reads height was used, therefore, the displacement

We replaced "therefore" with "and".

- p. 8, bottom, typo It reads testing the different as the

We corrected this sentence to: *"This makes the site ideal for testing the different maps, because the meteorological masts are impacted by both changes in tree height and forest edges."*

- p. 9, line 15, typo It reads had their roughness was set to 0.1 m

We removed the word "was" in this sentence.

- p. 9, below eq. 2 It reads the forest height was used to calculate a zero-plane displacement height d. I don't understand. Why is the displacement height related to the tree height?

The displacement height is often related the vegetation height. This is a widely used assumption and for further details we refer to, for example, Garratt (1992).

Please also note the supplement to this comment:
https://www.wind-energ-sci-discuss.net/wes-2018-10/wes-2018-10-AC1-supplement.pdf

**Supplement:**

[revised manuscript text omitted]

---

## Author Comment (AC2) · 24 Apr 2018

**1   Reponse Reviewer 2**

The manuscript does an excellent job at tackling the issues of data availability in land use maps for the calculation of wind resources in forested areas. The scientific significance and quality of the manuscript are both very high; I believe some sentences should be rewritten to be made clearer, and there are some minor inaccuracies with the use of English. I therefore recommend that after some minor technical corrections that manuscript be accepted.

*We thank the reviewer for the comments and have made modifications accordingly (see*

*below). The line numbers all refer to the track-changed manuscript where deletions are indicated in red and new text is indicated in blue.*

Some specific comments below.

- page 1, lines 10-13 (p1l10-13): While not unclear, sentence a bit convoluted and difficult to follow. Please rewrite.

We rewrote this to: "The improvements when using the ORA maps were both due to the higher roughness length and the higher resolution."

- p2l3: Please spell out WAsP and add website

Changed as suggested.

- p2l13: substitute colon between references (Jackson (1981); Raupach (1994)) with 'and'. p2l24: In a later study, Dellwik et al. (2006) demonstrated...

Changed as suggested.

- p3l1: add website for WindPRO

The website has been added and we reorganised the text, such that WindPRO is introduced earlier in the manuscript (lines p2l17-19).

- p3l7: ..impact of adding d to terrain data on model predictions.

Changed as suggested.
- p3l18: . . . about forest characteristics. . .

Changed as suggested.

- p3l19: During the last decade, ALS. . .

Changed as suggested.

- p3l26: . . . based on standard land-use. . .

Changed as suggested.

- p3l30: . . . how WAsP handled. . . . . . the speed of WAsP calculation.

We used "the" here to be consistent with the meaning of the acronym (Program), but perhaps this was confusing. We removed "the" in front of WAsP throughout the manuscript.

- p3l34-35: replace'(Sec. 2) with 'Section 2 (i.e. no brackets, spell out Section). Also spell out Section in Section 3.

This is done because of the journal requirements to formatting.

Sections: The headings of all sections, including introduction, results, discussions or summary must be numbered. Three levels of sectioning are allowed, e.g. 3, 3.1, and 3.1.1. The abbreviation "Sect." should be used when it appears in running text and should be followed by a number unless it comes at the beginning of a sentence.

- p4l5-8: Sentence not unclear but long and convoluted. Please rewrite.

We rewrote and split this sentence in two lines.

- p4l24: Section 5.

See our answer two comments back.

- p6l9:frequently

This has been corrected.

- p6l15: Why Carroll et al., 2010 while in Table 2 it's DiMiceli et al., (2011)?

The reference from DiMiceli points to the exact URL where the data can be downloaded, whereas the reference to Carroll specifically is about the way the data has been created.

- p6l7-16: Please add survey dates for the 4 land-use datasets.

These are already given in Table 2 in the column satellite coverage dates. We do not feel that there would be a benefit from adding the periods again in the text.

- p6l17: WindPRO

Changed as suggested.

- p6l18: Table 3 (i.e. capitalise 'Table')

Changed as suggested.

- p6-Table 2: Suggested new title: 'Summary of the different land-use data sources used for creating the roughness maps' Capitalise all letters in CORINE; perhaps also add that it's European data. Capitalise all letters in MODIS.

The table caption has been changed and all data sets have been capitalised.

- p7-Table 3: Suggested new title: 'Land use category and prescribed values of roughness length for each land-use dataset' CORINE section.  Last line: Broadleaved.  What is complex cultivation under CORINE? Maybe add simple explanation to table.  GLOBCOVER section.  What is 'pen' mixed broadleaved? Global Land Cover section. First line: substitute 'needleleaf' with 'needleleaved'

All these errors/suggestions have been applied.

- p8l16: Figure 1 shows the tree heights (right panel) and elevation (left panel) derived. . .

The references to the panels were added.

- p8l18: This makes the site ideal for testing the difference. . .  Please also state what difference.

We changed this line to: "This makes the site ideal for testing the different maps, because the flow at meteorological masts is impacted by both changes in roughness and the geostrophic roughness (see Sect. 1)."

- p8-Figure 1: Please use same ticks in axes X and Y. Also please specify that the units for elevation and tree heights are in m.

The figure now has the same ticks on both axes and the units are given above the color bar.

- p9l15: . . . and had their roughness set to 0.1m. . .

This was corrected.

- p9l16: m (i.e. not mm)

Changed to 0.0001 m.

- p9l24: 3.2.3 Changing map resolution

Changed as suggested.

- p9l25: To investigate the impact of map resolution. . .

Changed as suggested.

- p10l6: . . . file for compatibility with WAsP. . .

The word "the" is needed here.

- p10l8: Maybe add example in brackets at end of sentence, e.g.: '(e.g. ORA20) ?

Changed as suggested.

- p10l12-13: I would enclose in brackets the sentence: 'ORA maps that use. . . with and additional 'D'.

We added a full-stop after the first part of the sentence.

- p10l17: . . . described above needed to. . .

Changed as suggested.

- p10-Table 4. Suggested new title: Displacement height (m) determined from pixels. . .

Changed as suggested.

- p11l7: as for reference in page 2, substitute colon between references (Troen and De Baas (1987); Troen and Petersen (1989)) with 'and'.

Changed as suggested. Actually we meant to refer to Troen (1990) so the first of the references above has been changed.

- p11l10-11: 'This modelling chain is extensively described in Troen and Petersen (1989). Here we briefly summarise the steps:'

Changed as suggested.

- p11l21: Please remove comma after roughness lengths

Changed as suggested.

- p11l28: 'This is done by creating distance-weighted...

Changed as suggested.

- p12l1: Please briefly mention what the nodes are and what they represent.

We rewrote this sentence to make it more clear that we are talking about items in an array. Also we have added a reference where the algorithm is described in more detail.

- p12l4: as for references in page 2 and page 17, substitute colon between references (Sempreviva et al., (1990); Floors et al. (2011)) with 'and'.

Changed as suggested.

- p12l20: '...six of the seven masts, but at mast 6...'

Changed as suggested.

- p12-eq.6: I take A is the scale parameter of the Weibull distribution. Please make that explicit in the eq. description.

Yes, the symbol A is already defined at p11l26 and therefore it is not given here.

• p13l12: Please spell out Section

This has been changed to "Sect.", see journal guidelines in the answer to comment number 10.

• p13l17-18: Sentence not unclear but needs to be improved. Also please conjugate in plural those verbs with data as subject (e.g. data have, not data has)

We completely rewrote this sentence because it was indeed not clear what was meant. " These features can be seen more clearly in the histograms of the roughness lengths for each map (Fig. 3). For example, the roughness lengths of the grid cells in the ORA maps are higher on average and are spread over more bins than the satellite-based maps."

• p13l23: Please spell out Section

See answer to comment 10.

• p13l24: too many brackets for Fig.4

This has been corrected.

• p13l26: is filtering the QC-filtering? Maybe worth specifying?

We added QC.

• p13l27: Please put "<0.2" in brackets, not between commas.

Changed as suggested.

- p13l28: please remove 'flagged'

Changed as suggested.

- p14-Figure2. Suggested new title: Roughness maps from land-use datasets, and from ORA at 100m and 500m resolutions, colored by roughness values. Open circles...

We changed this to: "Roughness maps from land-use datasets, and from ORA at 100 and 500 m resolutions, colored by ... "

- p15. Figure 4: please remove 'the' before WAsP.

Corrected.

- p17l12-13: Not unclear but should be made clearer.

We added an extra sentence to clarify the evaluation strategy and refer back to Sect. where it is described in more detail. "For each pair of observed histograms, the histogram at the source location was used to predict the wind distribution at the target location and compared to the observed histogram at target location (see Sect. 4)."

- p17-Figure6. Suggested new title:

a) Mean absolute error in wind speed (%) and b) power density...

Changed as suggested.

- p18l18: Please spell out Section

See comment 10.

- p18l29: 'This shows that resolution is important...'

Changed as suggested.

- p19l8: 'The displacement height...' p19l16: '..investigations of model behavior'.

This error has been corrected.

- p19-Figure 7. The title describes the graph as mean absolute error, while the Y axis on the graph reads 'Relative error'. Please correct accordingly.

We modified the caption (see new caption Fig. 7).

- p20-Figures 8 and 9: Please remove 'The' at the beginning of titles.

This has been changed throughout the manuscript.

p21l5: '...into two types. Horizontal predictions ...'

Changed as suggested.

- p21l16: remove space between 41 and %

This is done in accordance with the journal policy: Spaces must be included between number and unit (e.g. 1 %, 1 m).

- p22l10: Should it be RMSE rather than just RMS?

No, because the error in this case is written as $\delta U$.

- p22l17: '... different descriptions of land use.'

Changed as suggested.

- p23l20-21: '... products with more detailed, but incorrect information (e.g. GLCC1000 vs GLOB300) when comparing only the commercially available maps.'

Corrected.

- p23l23: 'As argued by Jackson (1981) and Raupach (1994), forest density and forest height are likely to influence the optimal...'

Changed as suggested.

- p23l24-25: '... Plant Area Index (PAI) can also be estimated from airborne data...'

Changed as suggested.

- p24l4: '. . . ORA maps was tree height, whereas. . . '

Changed as suggested.

- p24l7: '. . . problematic since - in reality - the presence of clearings. . .
- p24l8: '..the overall turbulence levels and thereby increase the. . . '

Changed as suggested.

- p24l26-27: 'updated information on forest height and structure when evaluating. . . '
- p25l18: '. . . appropriate site data that describe forest height, type, and density.'

Changed as suggested.

- p25l19: I don't understand why spruces and pines are brought into the paper so abruptly (and at the very end of the manuscript!). Is it essential to talk about spruces and pines at this point?

This was indeed a bit out of place. We introduced the word 'coniferous' earlier in the conclusions instead (line p25l12).

- p25l24: '. . . DTU Wind Energy, where WAsP is maintained, developed, and sold.'

Changed as suggested.

- p26: Check positioning of brackets in Acknowledgements: there seems to be a mistake: bracket required after '(Grant Number 6172-00004B', not at the end of the sentence.

Corrected.

- References: Is it correct that he publication year is at the end? Please check the journal's suggested citation standard (I could not find it)

The journal formatting indeed says it should be in the end, see here: https://www.wind-energy-science.net/Copernicus_Publications_Reference_Types.pdf

Please also note the supplement to this comment:
https://www.wind-energ-sci-discuss.net/wes-2018-10/wes-2018-10-AC2-supplement.pdf

**Supplement:**

[revised manuscript text omitted]